# Harnessing speckle for a sub-femtometre resolved broadband wavemeter and laser stabilization

Nikolaus Klaus Metzger[1], Roman Spesyvtsev[1], Graham D. Bruce[1], Bill Miller[2], Gareth T. Maker[2], Graeme Malcolm[2], Michael Mazilu[1] & Kishan Dholakia[1]

The accurate determination and control of the wavelength of light is fundamental to many fields of science. Speckle patterns resulting from the interference of multiple reflections in disordered media are well-known to scramble the information content of light by complex but linear processes. However, these patterns are, in fact, exceptionally rich in information about the illuminating source. We use a fibre-coupled integrating sphere to generate wavelength-dependent speckle patterns, in combination with algorithms based on the transmission matrix method and principal component analysis, to realize a broadband and sensitive wavemeter. We demonstrate sub-femtometre wavelength resolution at a centre wavelength of 780 nm, and a broad calibrated measurement range from 488 to 1,064 nm. This compares favourably to the performance of conventional wavemeters. Using this speckle wavemeter as part of a feedback loop, we stabilize a 780 nm diode laser to achieve a linewidth better than 1 MHz.

[1] SUPA, School of Physics and Astronomy, University of St Andrews, Scotland KY16 9SS, UK. [2] M Squared Lasers Ltd, Venture Building, 1 Kelvin Campus, West of Scotland Science Park, Glasgow G20 0SP, UK. Correspondence and requests for materials should be addressed to K.D. (email: kd1@st-andrews.ac.uk).

ight propagation in disordered media is often regarded as a randomization process by which the information contained within an optical field is destroyed, and the speckle pattern that results from repeated scattering and interference is commonly deemed detrimental to an optical system. However, although highly complex, multiple scattering is nonetheless a linear, deterministic and reversible process[1]. A coherent beam propagating in a disordered medium yields a unique speckle pattern that depends on the spatial and temporal characteristics of the incident light field. In the spatial domain[2,3], light scattering in complex (disordered) media can be harnessed for imaging at depth[4], novel endoscopy[5] and even sub-wavelength focusing[6]. In a time-invariant complex medium, the temporal variations of the speckle pattern can be used to identify characteristics of the incident light, in particular as a spectrometer or wavemeter[7–20], with state-of-the-art demonstrations of wavelength resolution at the picometre level and operating ranges of over $1\,\mu m$ covering the visible and near-infrared spectrum.

In this letter, we use an integrating sphere as the random medium for wavelength detection. This acts like a highly sensitive and complex interferometer due to internal path-length differences on the metre scale, delivering a different speckle pattern for each wavelength in a system with a small footprint. The resultant wavemeter is demonstrated over a wide spectral operating range of $\sim 600\,nm$ in the visible and near-infrared spectrum, with wavelength resolution at the sub-femtometre level (0.3 fm measured at a centre wavelength of 780 nm). This resolution is up to two orders of magnitude better than all previous spectrometers using complex media[12,16,21]. Through the subsequent use of appropriate electronic feedback, we realize an alternative approach for laser frequency stabilization. We are able to stabilize an external cavity diode laser (ECDL) over 10 s, (a timescale suitable for, for example, laser cooling of rubidium atoms) and observe linewidth narrowing of the ECDL by a factor of four. Since the speckle-based stabilization scheme is not based on atomic absorption, it has the advantage of being able to lock at an arbitrary wavelength, as with Fabry-Perot-based stabilization schemes[22].

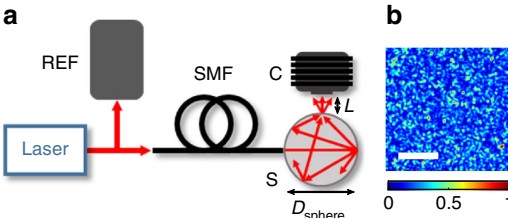

**Figure 1 | Experimental setup. (a)** Laser light is delivered into an integrating sphere (S) of diameter $D_{sphere}$, via an angled single mode fibre (SMF), where it undergoes numerous scattering events. A camera (C) located at a distance $L$ from the integrating sphere observation port captures the speckle patterns. A wavemeter or Fabry–Perot interferometer (REF) is used to benchmark the speckle wavemeter. **(b)** An example of a speckle pattern captured by the camera at a wavelength of 780 nm with an image size of $512 \times 512$ pixels (with individual pixels of $4.5 \times 4.5\,\mu m^2$). The white scale bar denotes 200 pixels, and the colour bar shows the intensity in normalized units.

## Results

**Wavemeter setup.** All data in this letter were acquired with the setup shown in Fig. 1. The laser beam is split into two paths. The first path is used for reference measurements of the laser wavelength with a commercial wavemeter or a Fabry–Perot interferometer. The second path is coupled into a Spectralon-coated integrating sphere through an angle-cleaved single-mode fibre to ensure a spatially-static input beam and eradicate back reflections, which could otherwise hinder stable laser operation. The speckle pattern produced inside the sphere is measured by a camera placed at a distance $L$ from the exit port of the integrating sphere. In contrast to interferometer-based wavemeters, no accurate alignment of the individual components is necessary. For

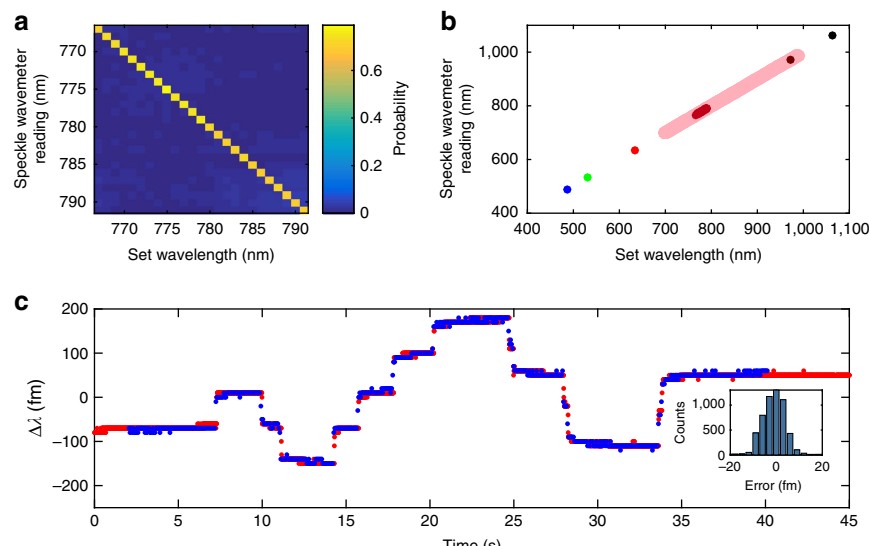

**Figure 2 | Wavelength measurement with the speckle wavemeter. (a)** Confusion matrix showing the probabilistic measurement of different set wavelengths of an external cavity diode laser, analysed with the transmission matrix method. **(b)** Measured wavelength using the same external cavity diode laser as in **a** (dark red), plus a broadly tunable titanium-sapphire laser (pink) and assorted other single-wavelength sources, across the broad operational range of 488–1,064 nm. **(c)** Principal component analysis is used to determine the wavelength with sub-picometre accuracy. The graphs show the measured wavelength of a modulated external cavity diode laser, obtained both from our speckle pattern reconstruction (blue) and a reference commercial wavemeter (red). Here, $\Delta\lambda = 0$ corresponds to a wavelength of 779.9431 nm. The inset shows the error distribution of the measurement with respect to the reference wavemeter with a s.d. of 7.5 fm. The data were taken within 10 min of the initial wavelength calibration.

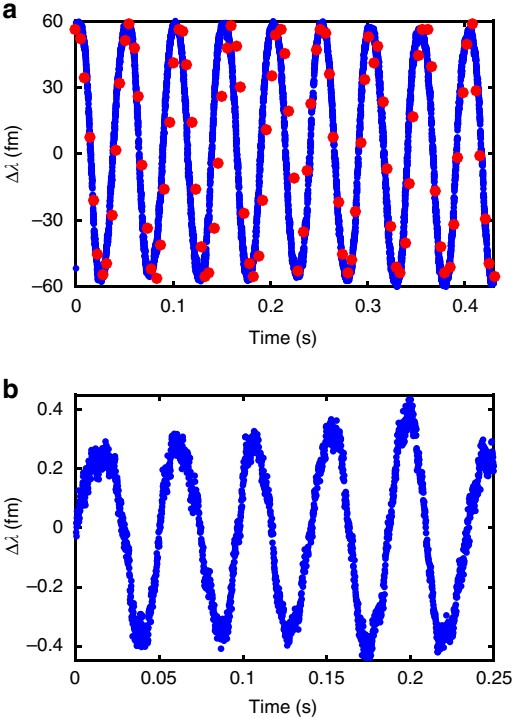

**Figure 3 | Resolution of the speckle wavemeter.** Wavelength modulation of an external cavity diode laser at $f = 21$ Hz and an amplitude of (**a**) 60 fm and (**b**) 0.3 fm, measured with the speckle wavemeter (blue) and the reference wavemeter (red). $\Delta\lambda = 0$ corresponds to an absolute wavelength of 780.2437 nm. The modulation at 0.3 fm is below the resolution of commercial wavemeters.

example, the camera and fibre can be offset by several degrees without perturbing the measurement results, as long as stability is maintained after the calibration step.

The mean free path in the sphere plays a prominent role in increasing the resolution of the system, analogous to the spectral to spatial mapping principle of a spectrometer. Thus, both the sphere diameter $D_{sphere}$ and propagation distance $L$ from sphere to camera dictate the resolution and range of the wavemeter, as detailed in the Methods section. Unless otherwise stated, here $D_{sphere} = 5$ cm and $L = 20$ cm. The acquisition rate of the wavemeter depends on the digital processing and the camera speed, which can be above 200 kHz for a small detector area. The total power sent to the sphere was below 1 mW for the highest frame rates, and can be lowered when a short exposure time and high frame rate are not required. Further details of the experimental setup can be found in the Methods section.

**Detection algorithm.** The core of the speckle wavemeter is the detection algorithm used to associate wavelength changes with the speckle pattern variations. To achieve a broad operating range, a coarse analysis using the transmission matrix method (TMM) follows standard approaches[7–13] for wavelength determination with a nanometre resolution, as shown in Fig. 2a,b. The intensity distribution recorded by the camera is $I = T \times S$, where $S$ is the input spectrum and $T$ the transmission matrix. This is constructed in a calibration stage, from speckle patterns recorded using established and well-known laser sources across the required wavelength range with a step of 1 nm. Subsequently, the unknown spectrum of incident light can be reconstructed from a speckle pattern, at the resolution of 1 nm, by performing left-inversion of the transmission matrix. However, noise can corrupt the reconstruction, we therefore use the truncated

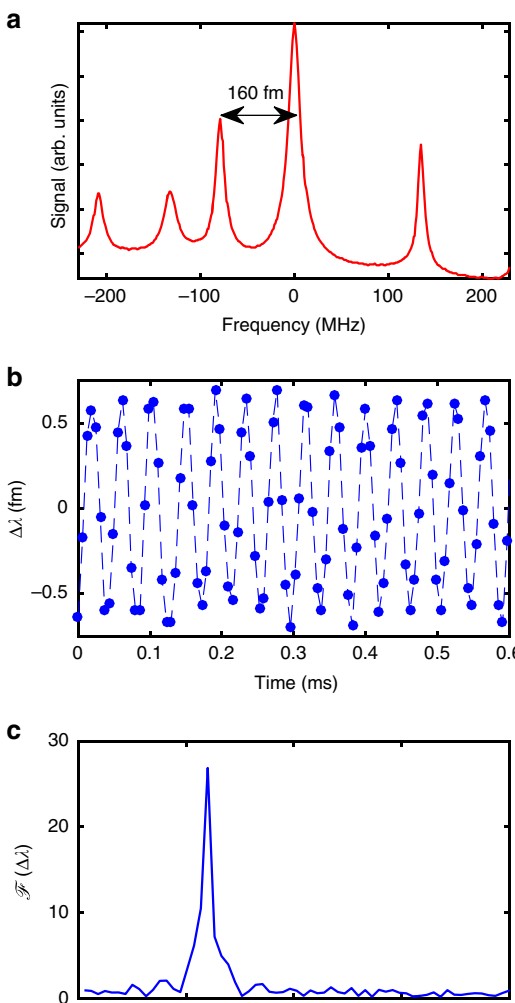

**Figure 4 | Measurement of laser dither tone.** (**a**) The hyperfine structure of rubidium, with the 160 fm separation of two crossover peaks indicated. The laser is locked to the largest peak at a dither frequency of 24 kHz with an amplitude of 1.3 fm. (**b**) Measurement of the dither tone while the laser is locked. To achieve a high sampling rate the image size is reduced to almost a line with 2,048 × 4 pixels. The points show individual acquisitions while the dashed line provides a guide to the eye. (**c**) Single-sided amplitude spectrum retrieving the lock dither tone frequency as 23.9(7) kHz, where the error is the half-width at half maximum of the peak.

singular value decomposition to find a pseudo inverse transmission matrix, as shown in ref. 11. At this stage, a nonlinear search algorithm[8–13] could be employed to further lower the noise in the retrieval of the spectrum and enable characterization of broadband light in a spectrometer configuration[7,13]. This part of our algorithm is exclusively used to determine the wavelength of an unknown light field with a precision of 1 nm, as shown in Fig. 2a,b and no further attempt has been made to retrieve a broad spectrum or increase the resolution. The demonstrated operational range of 488–1,064 nm is currently limited by the spectral response of the camera used.

To achieve a high resolution we use a different classification algorithm capable of accurately distinguishing small speckle changes caused by wavelength changes below 1 nm. This algorithm is based on principal component analysis (PCA), and is further detailed in the Methods section. Principal component analysis (or singular value decomposition) is an established tool

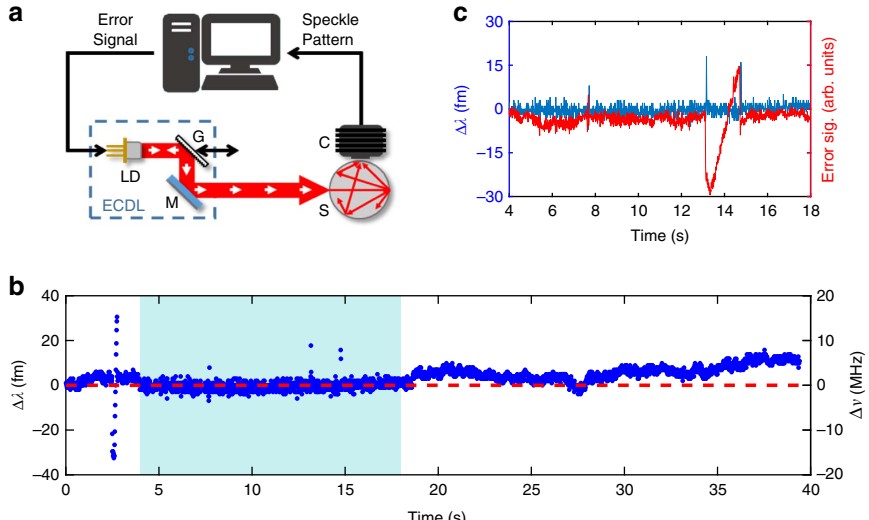

**Figure 5 | Laser stabilization.** (**a**) Experimental setup of the control loop: laser light is generated using an external cavity diode laser (ECDL) comprising laser diode (LD), diffraction grating (G) and a mirror (M). The integrating sphere wavemeter (sphere, S and camera, C) is employed as a detector of wavelength perturbations. The speckle image is analysed and a control algorithm hosted on a desktop computer generates an error signal which is used to regulate the laser wavelength by adjusting the DC current to the laser diode (LD). A controllable perturbation to the wavelength is applied by adjusting the cavity length (double-arrow). (**b**) Wavelength trace of the experiment: Initially the system is calibrated by linearly changing the drive current, defining the capture range of the speckle lock to ± 30 fm or ± 15 MHz at a centre wavelength of 780.24 nm. The midpoint of the calibration range is the wavelength setpoint (red dashed line). The laser is locked between 4 and 18 s (shaded region) and the linewidth of the laser narrows from 3.2 to 0.82 MHz. (**c**) Zoom of the lock period. We apply a perturbation via the grating, indicated by the error signal in red. The wavemeter output (blue) shows deviations only at points of rapid change outside the response time of the lock.

to decorrelate multivariate data[23,24]. In the field of optics PCA has, for example, been applied to wavelength detection[16], detector noise analysis[25], to distinguish surface scattering processes[26] and is commonly used in the spectral band recognition of mixed colourants[27], as well as in Raman analysis[28]. For our wavelength meter we project the data set (speckle images) onto its principal components and thereby retrieve the underlying wavelength modulation[16]. An initial calibration data set is acquired and used to correlate the speckle patterns to wavelength, and sets the measurement range of the high-resolution wavemeter. The density of the calibration set aids the accuracy of the wavemeter[16]. Subsequently, all measurements are projected against this calibration set to extract the unknown wavelength. Importantly, the wavelength-dependent changes in the speckle pattern that are captured by PCA are subtle in comparison to the TMM. A comparison of both approaches is shown in the Methods section. An example of the retrieved wavelength trace (blue) is shown in Fig. 2c together with the reference wavemeter reading (red). We note that, in order to measure an unknown wavelength to high precision, these two algorithms can be used sequentially. All subsequent measurements in this paper were performed close to 780 nm, but could in principle be performed at any wavelength within the operational range of the camera.

**Wavemeter resolution.** After the calibration of the desired operating range, we test our speckle wavemeter by measuring the change in wavelength from an ECDL centred at 780.2437 nm. We apply a controlled wavelength modulation by varying the current to the laser diode at 21 Hz, as shown in Fig. 3.

For the case of relatively large wavelength modulation of 60 fm, Fig. 3a shows the good agreement between the wavelength measured using the speckle wavemeter, where we set $L = 2$ cm, and a reference wavemeter. To explore the resolution limit of our approach, we increase the distance to the camera to $L = 20$ cm and decrease the wavelength modulation amplitude by a factor of

200. These measurements are below the wavelength resolution of our commercial wavemeter, but interpolating over five runs gives agreement within 10% of interpolated Fabry–Perot interferometer measurements of the wavelength. In Fig. 3 the centre wavelength has been subtracted from the speckle wavemeter wavelength readings and displayed as $\Delta\lambda$ in femtometres, where $\Delta\lambda = 0$ corresponds to an absolute wavelength of 780.2437 nm. Clearly, with a high-resolution commercial wavemeter a more accurate calibration would be possible and an absolute wavelength reading could be presented. Further reduction of the modulation amplitude was not possible due to the limited resolution of the modulation signal source. A more detailed analysis of the range and resolution of the speckle wavemeter and its dependence on experimental design parameters is presented in the Methods section.

Using the speckle wavemeter as a detector within a calibrated rubidium absorption spectroscopy setup, we also resolve the wavelength changes of a stabilized titanium-sapphire laser (M Squared Lasers Ltd SolsTiS). The laser is stabilized by top-of-fringe locking[29] to the crossover between the $F = 2 \rightarrow F' = 2$ and $F = 2 \rightarrow F' = 3$ hyperfine transitions in the D2-line of $^{87}$Rb (see Fig. 4a). Importantly, this locking technique uses a dither tone at a frequency of 24 kHz with an amplitude of 1.3 fm). This modulation signal was wavelength calibrated, and is utilized as a calibration standard for our wavemeter. The speckle wavemeter is able to correctly resolve this dither tone, as shown in Fig. 4b,c.

A detailed comparison of our wavemeter to current commercial and research-grade wavemeters and spectrometers can be found in Supplementary Table 2.

**Laser stabilization.** Having bench tested our wavelength meter, we also incorporate it within the wavelength stabilization control loop of the ECDL, as depicted in Fig. 5a. Here, the speckle wavemeter is once again used as a detector: the speckle image is analysed using the PCA algorithm to determine the emitted wavelength. A subsequent proportional-integral-derivative

algorithm calculates an error signal from the measured wavelength to a predefined wavelength setpoint. The error signal regulates the ECDL diode drive current and thus the laser wavelength. Initially the wavemeter is calibrated across $\pm 30$ fm ($\pm 15$ MHz) from the stabilization setpoint, which sets the capture range of the speckle lock. The grating of the ECDL is employed as an external controllable wavelength perturbation, which we use to demonstrate the stabilization performance of the speckle lock.

Figure 5b shows a wavelength trace captured by the external wavemeter during the stabilization. The experiment starts at 2.5 s with the 60 fm-wide calibration ramp, then the laser is locked to the central speckle image (at 780.24 nm) from 4 to 18 s. From 18 s onwards the ECDL is free running without feedback control. During stabilization we achieve a linewidth narrowing from 3.2 MHz (rms) to 0.82 MHz (rms) of the ECDL over a 10 s observation time (for both measurements), which is ample for applications such as the efficient laser cooling of atoms.

As shown in Fig. 5c, between 13 and 15 s we apply a sawtooth modulation to the ECDL grating which corresponds to a 50 fm variation of the unstabilized laser wavelength. The resultant error signal shows the compensation applied to the locked laser and the wavemeter only detects wavelength changes at the start and end of the ramp, when the perturbation is largest. Increasing the update rate of the control loop, which is currently limited to 200 Hz, would minimize the amplitude of these spikes.

## Discussion

This work presents a practical speckle-based wavemeter with sub-femtometre resolution, which retains a broad operating range. A central element of the operating principle of the device is the use of an integrating sphere to create speckle patterns with a high sensitivity to wavelength changes through a compact and near lossless propagation path. A two-step algorithm based on the transmission matrix method and principal component analysis was applied to resolve wavelengths within the range 488–1,064 nm and changes in wavelength below 0.3 fm respectively. A full specification of the present performance of the wavemeter is presented in Supplementary Table 1.

In comparison to the use of an integrating sphere, fibre-based[9,12,13] speckle wavemeters have realized picometre resolution to date. Progress towards attometre-level resolution of our wavemeter is currently limited by our 16-bit amplitude modulation source and the availability of an accurate wavelength reference to calibrate the variability of the principal components. In theory, the selectivity of the integrating sphere coating (Spectralon) supports an operational range from 0.2 to 2.5 μm at high efficiency (reflectivity $> 90\%$). In practice, this operational range was limited by the response of the CMOS chip of our cameras.

Our present system is limited in its long-term stability (Supplementary Note 1; Supplementary Fig. 1), but mechanical and thermal stabilization engineering could form part of a future study. Alternatively, a regular recalibration step, as can be found in many commercial wavemeters, can be used to maintain accuracy over longer periods of time.

The speckle-based wavemeter was also used as part of a laser stabilization setup. Encouragingly, the system achieved laser stability over several seconds without any thermal or vibration management, and reduced the linewidth of a diode laser by a factor of four. It is important to note that the frequency stability of the speckle-locked laser is presently not compared to any reference laser, and thus the measurement was limited to the accuracy of the wavelength meter. The demonstrated capture range of 60 fm (corresponding to 30 MHz at 780 nm) is currently

small compared to other approaches such as polarization spectroscopy[30], which can achieve gigahertz capture range for lasers with a comparable linewidth to those discussed here.

The locking technique described here could aid the development of portable atom-based quantum technologies[31] due to the small footprint and insensitivity to misalignment. The speckle-based stabilization is a convenient method to lock at arbitrary detuning from an atomic resonance, such as the tens of gigahertz typically required in atom interferometry experiments[32]. Furthermore, compared to dither locking of the laser to an atomic saturated absorption feature, speckle-based stabilization has the advantage of eliminating any frequency modulation of the laser. Finally, the extension of our approach to the simultaneous detection and measurement of multiple wavelengths[33] could also allow a single device to stabilize all of the various lasers (cooling, repumper, Raman and trapping) required in quantum technologies based on cold atoms.

## Methods

**Details of the experimental setup.** For the wavemeter study, several well-known and established laser systems were employed to determine the broadband range we were able to achieve with our approach. For the resolution and stabilization studies, an ECDL (Sacher Lasertechnik Lion TE-520, 780 nm and linewidth of 3 MHz over 10 s) was used as a wavelength tunable source. A commercial wavemeter (HighFinesse WS7, resolution 5 MHz) and a Fabry–Perot interferometer (Burleigh, free spectral range 2 GHz, finesse 200, ~10 MHz linewidth resolution) were used for wavelength reference to benchmark our system. The speckle patterns were captured using a xiQ—USB3 Vision Camera for the wavelength measurements. A fast camera with a maximum frame rate above 200 kHz (Mikrotron monochrome EoSens 4CXP, 4 megapixel CMOS camera with an Active Silicon FireBird CoaXPress frame grabber) was used for the laser stabilization experiments. For all measurements, the camera exposure and optical power were adjusted to avoid image saturation with the aid of an image histogram, as shown in Supplementary Fig. 2. The speckle patterns were subsequently analysed on a 2.4 GHz dual core personal computer.

**Speckle image analysis algorithm.** In the integrating sphere wavemeter every speckle pattern serves as a marker for a wavelength. An initial training phase is used to correlate training speckle patterns to wavelengths. In a second step every new acquired frame is projected against the initial training set and the current wavelength is retrieved. In the TMM algorithm, our training set is a series of speckle images at known wavelengths with a step size of 1 nm. The image pixels associated with each wavelength form the column vectors in our calibration matrix, that converts between spatial patterns and input frequencies. Figure 6a shows an example of a calibration matrix for wavelengths of 769–791 nm. The average correlation over the image pixels, shown in Fig. 6b, is close to zero for a wavelength shift of 1 nm.

An example training set that converts between spatial patterns and input frequencies at the femtometre level is shown in Fig. 6c. The spectral correlation function in Fig. 6d shows strong correlation over a range of at least 300 fm. Comparison of both calibration matrices in Fig. 6a,c makes it evident that at the femtometre scale the overall speckle variability has decreased significantly. Thus, to correlate the speckle pattern to wavelength variations of $<1$ nm, we have implemented the PCA approach using Matlab (and in C++ for the laser stabilization experiment) using the standard template library for linear algebra. The PCA is applied to speckle images $y_i$, which are acquired for a varying wavelength $\lambda$ (subscript $i$ denotes the pixel number). In Fig. 7 we plot the projections of the measured speckle patterns $y_i(\lambda)$ in PC space, $PCj = U_{ji} \cdot y_i$ where $U_{ji}$ corresponds to the principal components with index $j \leq 4$. The values on the $y$-axis are a relative marker of the variation as the variance of the speckle pattern is multiplied by the coordinate of the corresponding eigenvector $U$, with values between $-1$ and 1. The PCs for wavelength modulations of the ECDL at 60 fm (Fig. 7a) and 0.3 fm (Fig. 7b) are therefore represented relative to the sign of the eigenvector and can be inverted. (Here, the speckle image size is $256 \times 16$ pixels, the camera distance $L = 2$ cm and the integrating sphere diameter $D_{sphere} = 5$ cm for the 60 fm modulation and $D_{sphere} = 10$ cm for the 0.3 fm modulation.)

We observe that each PC shows an oscillatory behaviour that we further analyse by Fourier transforming (using the single-sided amplitude spectrum) the PC traces (Fig. 7c,d). We denote the Fourier transform of PC1 as $\mathscr{F}(PC1)$. We observe that PC1–4 are showing analogous oscillations. Here, PC1 is capturing wavelength modulation as generated by the laser diode drive current modulation, while PC2 detects the second order variations of the speckle pattern. Similarly, higher order PCs detect higher order speckle pattern variations that are each orthogonal and uncorrelated to each other. This orthogonality manifests itself also in the single-sided amplitude spectrum of the PCs time series, where we observe that PC3 and PC4 exhibit modulations corresponding to the third and fourth harmonic of the wavelength modulation frequency $f = 21$ Hz, as shown in Fig. 7c.

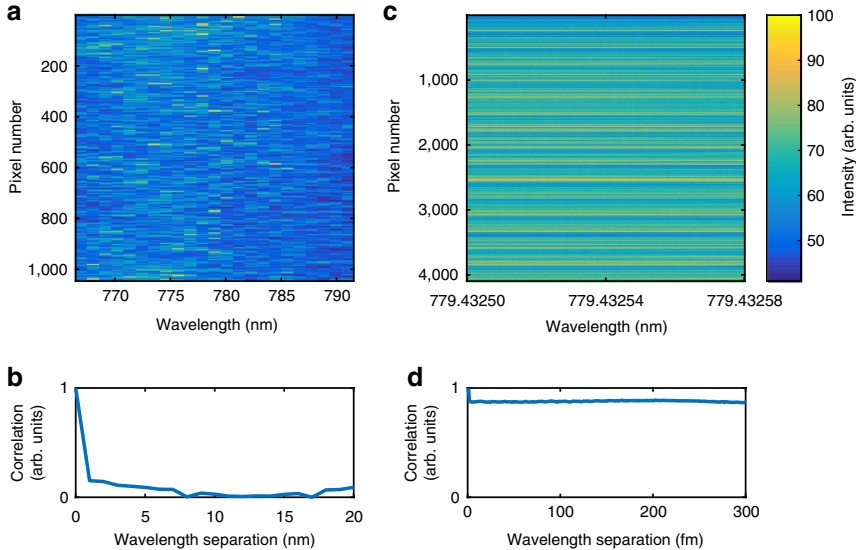

**Figure 6 | Comparison of transmission matrix method and principal component analysis.** (**a**) The transmission matrix method calibration matrix that converts between spatial patterns and input frequencies for wavelengths of 769–791 nm. The image pixels associated with each wavelength form the column vectors in the matrix. (**b**) Spectral correlation function of **a**. (**c**) Principal component analysis training set that converts between spatial patterns and input frequencies with a wavelength variability of 80 fm. (**d**) Spectral correlation function of **c**. The corresponding pixel intensity for **a**,**c** is shown in the right hand colour bar.

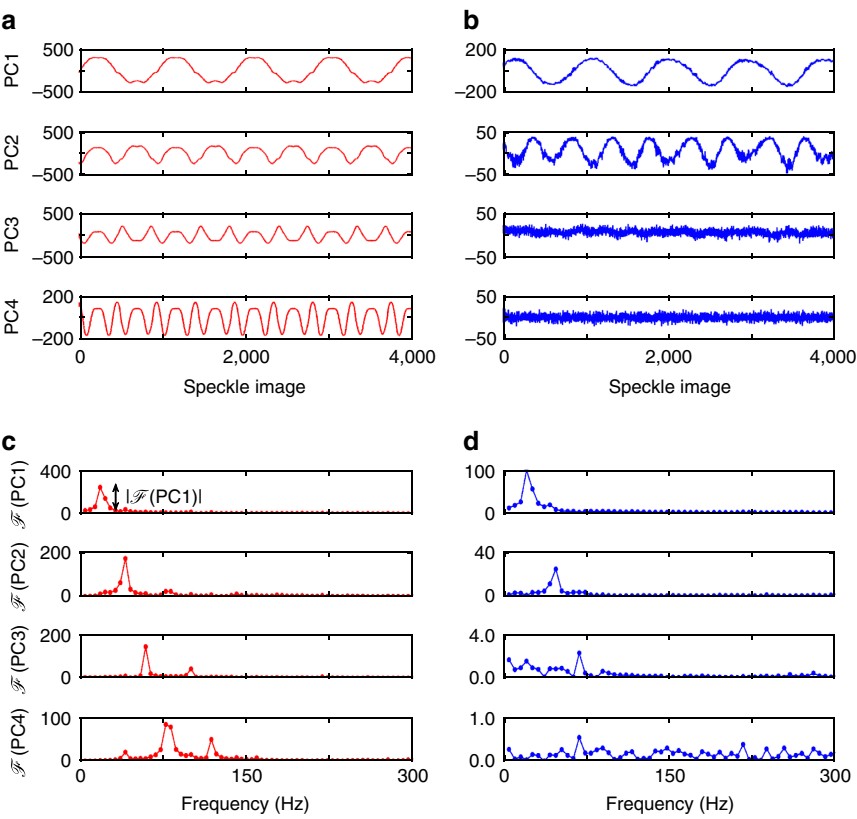

**Figure 7 | Principal component analysis.** Principal components (PC) 1–4 for a sinusoidal (at $f = 21$ Hz) wavelength modulation of (**a**) 60 fm and (**b**) 0.3 fm. (**c**,**d**) Single-sided amplitude spectrum obtained by Fourier transform of the PCs presented in **a**,**b** respectively. The peak amplitude $|\mathscr{F}(\mathrm{PC1})|$, which is used to extract the resolution and range of the analysis in Fig. 8, is highlighted in **c**.

**Measurement range and resolution.** The PCA algorithm is a powerful tool to detect small changes in the overall speckle pattern. In this section we investigate the range and resolution of the PCA solely.

One method to quantify the resolution and range limits of our wavemeter is to determine the amplitude of the modulation peak $|\mathscr{F}(\mathrm{PC1})|$ of the single-sided amplitude spectrum of the first PC, as indicated in Fig. 7c. The height of this peak correlates directly with the wavelength modulation and should follow a linear dependency. Deviations from this at low (high) wavelength modulation indicate an incorrect retrieval of the modulation and hence the resolution limit (range). This method has been experimentally verified for several setups as a method to quantify

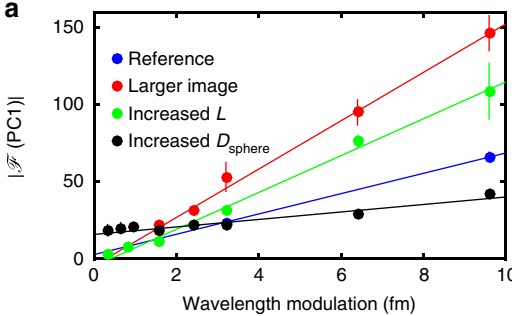

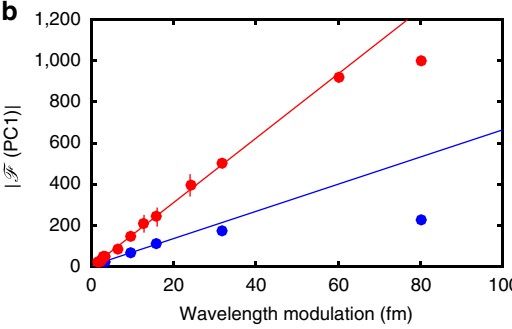

**Figure 8 | Optimization of range and resolution.** (**a**) Detected modulation peak $|\mathscr{F}(PC1)|$ as a function of wavelength modulation at $f = 21$ Hz. Error bars indicate the s.d. over five measurements. Different experimental parameters are investigated (see Table 1): Blue and red are studies with image sizes of $256 \times 16$ and $256 \times 128$ pixels, respectively. Blue is used as a reference and subsequent measurements were compared to the results. Green indicates an increased distance $L$ from 2 to 40 cm in comparison to blue. Black is a study with an increased diameter of the integrating sphere $D_{sphere}$ in comparison to blue: from 5 to 10 cm. (**b**) The bandwidth limit of the principal component analysis is the point at which the dependence of $|\mathscr{F}(PC1)|$ on wavelength modulation deviates from linearity. For two different speckle image sizes of $256 \times 16$ pixels (blue) and $256 \times 128$ pixels (red), this limit is 16 and 60 fm respectively. The bandwidth increases with increased image size.

**Table 1 | Optimization of range and resolution.**

| Image size (pixels) | $D_{sphere}$ (cm) | $L$ (cm) | Resolution (fm) | Range |
|---|---|---|---|---|
| 2,336 × 1,024 | 2.5 | 2 | 80 | 8 nm |
| 256 × 16 | 5 | 2 | 3.2 | 16 fm |
| 2,336 × 1,728 | 5 | 2 | 1.2 | 1 nm |
| 256 × 16 | 5 | 20 | ≤ 0.3 | 16 fm |
| 256 × 16 | 5 | 40 | ≤ 0.3 | 16 fm |
| 256 × 16 | 10 | 2 | ≤ 0.3 | 16 fm |

Resolution limit and measurement range of a speckle-based wavemeter using principal component analysis, measured at a centre wavelength of 780 nm.

The resolution of the method depends on the changes in the speckle pattern with respect to the incoming wavelength. When the wavelength changes, the speckle decorrelates. In previous work[7,12], the half-width at half maximum of the correlation function was used as an indicator for the ultimate resolution of the device. However, this method depends on the reconstruction algorithm and the ratio of the speckle variation to the detection noise[12]. Therefore, the actual resolution is determined by the ability to discern speckle patterns from two closely spaced spectral lines. In this work we are able to achieve resolution well below the half-width at half maximum of the speckle correlation. In the presented ECDL setup we are driving a piezo to tune the wavelength. To eliminate hysteresis effects due to the piezo, we used a sinusoidal modulation signal and the frequency retrieval as a marker for the resolution of the system.

**Data availability.** The data that support the findings of this study are available from the University of St Andrews Research Data Repository at doi:10.17630/9448553f-a806-464b-a41c-7e2f0c80522b.

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

the wavelength resolution. In the lower limit, this corresponds to sub-pixel changes in the speckle pattern, and in the upper limit, to changes occurring outside the finite image range. Figure 8 collates $|\mathscr{F}(PC1)|$ as a function of wavelength modulation amplitude. This is investigated for different experimental parameters (shown in Fig. 8) for small wavelength modulations of 0.3–10 fm. In future work, the theoretical model of our configuration which is presented in Supplementary Note 2 and Supplementary Figs 2 and 3 could guide the optimal design parameters of subsequent generations of our wavemeter.

A reference line (shown in blue) is measured for a $256 \times 16$ pixel image at $L = 2$ cm from an integrating sphere with $D_{sphere} = 5$ cm, which gives a resolution of 3.2 fm and range of 16 fm. First, we studied the influence of the total number of pixels. We observed that an eight times larger speckle image size (red line, $256 \times 128$ pixels) enhances the outcome of the PCA and decreases the resolution limit from 3.2 to 1.2 fm and increases the range to 60 fm. This can be explained by considering that a bigger detection array is able to capture smaller distributed perturbations.

Another important parameter is the propagation distance $L$ of the speckle pattern after the output from the sphere. Indeed, finer speckle structures can be resolved by the camera as this distance increases. The upper spatial frequency limit contained within a speckle pattern[34] is given by $\xi_{cutoff} = \frac{D_{port}}{\lambda L}$, where $D_{port}$ is the integrating sphere port diameter, $\lambda$ is the wavelength and $L$ the distance between the port and the camera detector array. For the detector array the Nyquist frequency can be calculated from half the element spacing[35,36] of 7 μm to 71.4 cycles mm$^{-1}$. To avoid aliasing, the maximum spatial frequency contained in the patterns should be equal to the spatial Nyquist frequency. With a port diameter of 1 cm the camera should be ~18 cm away from the sphere. Here, increased distances $L$ of 20 and 40 cm were investigated and resolution was found to increase for $L \geq 18$ cm (depicted for $L = 40$ cm in Fig. 8a, green) to 0.3 fm.

In addition, to enhance the spatial resolution of the system one can increase the mean free path within the sphere. When the sphere diameter is increased from 5 to 10 cm (depicted in Fig. 8a $D = 10$ cm in black) a resolution of 0.3 fm can be achieved. Table 1 summarizes the parameters in detail and shows the resolvable minimum wavelength modulation amplitude, as well as the range.

23. Shlens, J. A tutorial on principal component analysis. Preprint at http://arXiv.org/abs/1404.1100 (2014).
24. Brand, M. Fast low-rank modifications of the thin singular value decomposition. *Linear Algebra Appl.* **415,** 20–30 (2006).
25. Ferrero, A., Alda, J., Campos, J., López-Alonso, J. M. & Pons, A. Principal components analysis of the photoresponse nonuniformity of a matrix detector. *Appl. Opt.* **46,** 9–17 (2007).
26. Ferrero, A. *et al.* Principal components analysis on the spectral bidirectional reflectance distribution function of ceramic colour standards. *Opt. Express* **19,** 19199–19211 (2011).
27. Fairman, M. H. & Brill, H. S. The principal components of reflectances. *Color Res. Appl.* **29,** 104–110 (2004).
28. Jess, P. R. T. *et al.* Early detection of cervical neoplasia by Raman spectroscopy. *Int. J. Cancer* **121,** 2723–2728 (2007).
29. Hänsch, T. W., Levenson, M. D. & Schawlow, A. L. Complete hyperfine structure of a molecular iodine line. *Phys. Rev. Lett.* **26,** 946–949 (1971).
30. Pearman, C. P. *et al.* Polarization spectroscopy of a closed atomic transition: applications to laser frequency locking. *J. Phys. B* **35,** 5141–5151 (2002).
31. Bongs, K. *et al.* The UK National Quantum Technologies Hub in sensors and metrology. *Proc. SPIE Quantum Optics* **9900,** 990009 (2016).
32. Cronin, A. D., Schmiedmayer, J. & Pritchard, D. E. Optics and interferometry with atoms and molecules. *Rev. Mod. Phys.* **81,** 1051–1129 (2009).
33. Coluccelli, N. *et al.* The optical frequency comb fibre spectrometer. *Nat. Commun.* **7,** 12995 (2016).
34. Boreman, G. D., Centore, A. B. & Sun, Y. Generation of laser speckle with an integrating sphere. *Opt. Eng.* **29,** 339–342 (1990).
35. Pozo, A. M. & Rubiño, M. Comparative analysis of techniques for measuring the modulation transfer functions of charge-coupled devices based on the generation of laser speckle. *Appl. Opt.* **44,** 1543–1547 (2005).
36. Pozo, A. M., Ferrero, A., Rubiño, M., Campos, M. J. & Pons, A. Improvements for determining the modulation transfer function of charge-coupled devices by the speckle method. *Opt. Express* **14,** 5928–5936 (2006).

## Acknowledgements

N.K.M. acknowledges support from Dr C.T.A. Brown and thanks Dr T.J. Edwards and Dr S. Jiang for fruitful discussions. We acknowledge funding from EPSRC through an Impact Acceleration Account (IAA) Case Study award and grants EP/J01771X/1 and EP/M000869/1.

## Author contributions

N.K.M., M.M. and K.D. developed the project. N.K.M. developed the algorithms and conducted the experiments. N.K.M. and G.D.B. analysed the data. R.S. implemented the C++ real-time control algorithm, N.K.M. developed the C++ feedback control loop. B.M., G.T.M. and G.M. supported the work with a laser loan and technical advice. N.K.M., G.D.B. and K.D. wrote the paper with contributions from all the authors. K.D. supervised the study.
