## [Peer Review File · Nature Communications]

Reviewers' comments:

Reviewer #1 (Remarks to the Author):

Harnessing speckle patterns from an integrating sphere for a broadband, high resolution wavemeter and laser stabilization

N K Metzger, R Spesytysev, B Miller, G T Maker, G Malcolm, M Mazilu, K Dholakia

This important paper shows that the wavelength dependence of the speckle pattern produced using a fibre-coupled integrating sphere can be used to perform spectroscopy with sub-femtometre resolution by matching the imaged speckle to calibration patterns (transfer matrix method, TMM) and interpolating between them (principal component analysis, PCA). Although the basic principle is not new (see e.g. [10]), the 5cm diameter, Spectralon-coated sphere provides a far greater range of interfering path differences than previous attempts using multimode fibres [2, 10] etc. [7, 8], improving the spectral resolution by more than three orders of magnitude and considerably surpassing the best commercial spectrometers. This is a significant and fascinating result.

Precise wavelength measurement is necessary for the adjustment, characterization and stabilization of tunable lasers, and the spectroscopic characterization and identification of atomic and molecular species. Despite centuries of development, all conventional spectrometers comprise delicate assemblies of high-precision optical components and, in most cases, complex mechanisms that allow mechanical movement with sub-wavelength accuracy. The cheapest instruments cost hundreds or thousands of dollars, and most in everyday research use cost tens of thousands. The speckle-based technique requires stability of the integrating sphere and its optical fibre coupling, but is otherwise a robust arrangement; there are no moving parts, and the components are not fundamentally expensive. Since all of the light in principle reaches the detector, the intensity sensitivity and signal-to-noise are good; the two-dimensional imaging effectively over-samples the spectral information for experimental robustness and interpolated resolution enhancement; and, as the imaging arrays use the same technologies as conventional instruments, the measurement rate matches that of current devices. Significant computational processing is required, but computing power is nowadays cheap and plentiful. The St Andrews group has for the first time demonstrated that this elegant and sturdy device can exceed the performance of conventional spectrometers and wavemeters.

Users will wish to know the wavelength range, resolution, sensitivity and update rate of the wavemeter, and its ability as a spectrometer to distinguish between simultaneous spectral components. The St Andrews group has demonstrated the resolution and update rate, and that the operating wavelength range is in principle as wide as that of the imaging array, sphere reflectivity, and fibre delivery. In its prototype implementation, however, only a narrow wavelength range has been covered continuously, and one wavelength at a time. That preliminary findings should be published before the full performance has been determined is correct and important; but it would be helpful if the paper were blunter in the abstract and introduction about what has and has not been demonstrated, and if perhaps it offered some speculation about these aspects in the later stages.

There were similarly a few places in which the language and terminology of the paper could have been clearer or more precise. Detection is not the same as measurement; nor is versatility the same as performance. Without further explanation, the meaning of bandwidth is unclear in a device for which both the measurement rate and optical spectral range are of interest. A wavelength below one nanometre is different from a wavelength variation of less than a nanometre. There are other places where one is left unsure whether a point has been demonstrated or just asserted; whether the transfer matrix method (TMM) and principal component analysis (PCA) were used separately or combined; and, if the latter, how this was done. A critical reading of the manuscript would identify a number of places at which the phrasing could be improved to render it more accurate and unambiguous. Perhaps because of the journal's unfortunate habit of separating methods from results and principles, there were also places where the logical order seemed a little muddled: I suggest that the authors restrict the methods section to the less important technical details, and allow some of the more important ones (such as of the integrating sphere, interference path lengths, imaging camera, and their relative geometry) into the main text.

Nonetheless, this is scientifically a compelling paper. The demonstration is utterly convincing, the characterization thorough (especially given that, when out-performing conventional instruments, it is not easy), and the illustrations clear and attractive. The 10 second duration of the temporal stability at the highest resolution (p10) might have been mentioned earlier – it is unsurprising in the apparent absence of thermal stabilization, and an aspect that is more interesting than limiting. I have a minor

quibble, below, about the interpretation of Fig 2 of the Methods section, but the analysis is otherwise excellent.

This paper heralds a new family of spectroscopic devices of greater resolution, robustness and affordability than those which have been developed over decades and centuries. If the (already surprisingly good) instrument stability can be slightly improved, and the continuous operation range extended, it is likely to replace all current wavemeter designs; and, if concurrent detection of different wavelengths can be achieved (which seems likely – see e.g. Nature Comms 7, 12995 (2017)), it will also have a big impact for high-resolution spectroscopy. Even for laser wavelength stabilization, which is here reported to illustrate the device's performance, there will be many applications, for it is currently difficult to eliminate drift in lasers at wavelengths separated from reference lines by more than 10 GHz or so. To take two examples: the Raman transitions in atom interferometry must typically be detuned from single-photon resonance by 10-100 GHz, and a simple way of achieving this, with an accuracy of 10-100 MHz, would be popular amongst the emerging quantum technology community; if multi-wavelength operations were possible, a single device could monitor and stabilize the wavelengths of different lasers (e.g. in cold atom physics, the cooling, repumping, Raman and trapping lasers) simultaneously.

Aside from its technological and scientific implications, this work prompts one to wonder how speckle patterns evolve with wavelength, their topology (see e.g. Berry & Dennis, Proc R Soc Lond A 456, 2059 (2000)), whether their spatial frequency components alone are of interest, how the performance is enhanced by the two-dimensional measurement*, how it will be affected when other spectral components are also present, and how the PCA capture range is determined by the apparatus parameters. While this is not the first paper to explore the phenomenon, the authors should not be shy to address, at least in passing, some of the more illuminating conceptual aspects and questions.

In summary, this is an important and thought-provoking result, scientifically sound and of broad interest, which unquestionably merits publication in Nature Communications. It would benefit, though, from some textual polishing and re-ordering.

Some detailed suggestions follow overleaf. It is a pity that the manuscript draft lacks line numbering.

Tim Freearde, Southampton

* e.g. similar to the 'visual microphone' <http://people.csail.mit.edu/mrub/VisualMic/>
<https://www.youtube.com/watch?v=FKXOucXB4a8>

title	this could be snappier, and might for example mention the sub-fm performance
abstract, line 3	'original' here is ambiguous; perhaps 'novel'
abstract, line 15	the laser is stabilized to the wavelength of the Rb line; there is no Rb in the control loop
p1 para 1 line 1	'in a disordered medium', or 'in disordered media'
p2 para 1 line 1	'in such media'
p2 para 1 line 3	can speckle patterns (as opposed to a scientific field or community) possess a paradigm? Indeed, is it so rare for a characteristic to be both a pro and a con?
p2 para 2 line 6	'a smaller footprint'; is 'retain ... information' really correct?
p2 para 3 line 4	perhaps replace 'such as mode identification' with e.g. 'such as its transverse mode'
p2 para 4 line 2	'wavelength measurement' or 'wavelength determination'
p2 para 4 line 5	it would be interesting to learn how 'exceptionally long' is the interfering path difference
p2 para 4 line 8	'two orders of magnitude better'
p2 para 4 line 10	it would be helpful to distinguish throughout between signal bandwidth, (optical) spectral bandwidth and spectral operating range
p2 para 4 line 11	'visible and near-infrared spectrum'
p2 para 5 line 2	'laser frequency fluctuations'
p2 para 5 line 3	'use of appropriate electronic feedback'
p2 para 5 line 6	'absorption, as are Fabry-Perot based'
p2 para 5 line 7	'the capture range...' sentence is partly repeated; omit first version
p3 para 1 line 2	omit comma after 'sphere'
p3 para 1 line 4	were the lasers really evaluated, or simply used?
p3 para 1	this paragraph is long and a little jumbled; it contains many technical details (e.g. ECDL, F-P interferometer, frame grabber, computer processor) that are irrelevant to the thrust of the paper and would sit better in the 'methods' section, but omits the characteristics of the integrating sphere.
p3 para 1 line 31	given that the speckle pattern results from the interference between different 'trajectories', perhaps some hint of this process could be given here, and the 'trajectory' description emphasized less?
p3 para 1 line 37	'one would require a 300 km long fibre' (note that the spelling of 'fibre'/'fiber' is inconsistent in this paper)
p4 para 1 line 6	what is meant by 'a high degree of disorder'? Has this been quantified/measured?
p4 para 1 line 8	one could begin a new paragraph at 'The core of...'
p4 para 1 line 10	omit 'out'
p4 para 1 line 14	what is meant by 'wavelength determination above one nanometer'? Perhaps 'wavelength determination for variations above one nanometer'?
p4 para 1 line 17	the calibration process would benefit from further explanation. The laser sources are perhaps irrelevant: several common laser lines were used.
p4 para 1 line 21	the 'truncated singular value decomposition' might deserve a reference for the uninitiated
p4 para 1 line 29	is the operational range limited to 488-1064 nm, or is this just the range over which it has been demonstrated?
p5 Fig 2 caption	it would be helpful to explain the abbreviations TMM, PCA and ECDL within the caption. The resolution demonstrated in (b) appears to be rather better than the 1 nm stated.
p5 para 1 line 8	are the PCA and TM methods used together?
p5 para 1 line 9	perhaps 'Principal component analysis (or singular value decomposition (SVD)) is an established tool', or 'Principal component analysis and singular value decomposition (SVD) are established tools'
p6 para 1	it would be interesting to know how the wavelength accuracy varies with its proximity to the calibration wavelengths
p6 para 2	this paragraph is unclear, and becomes partly intelligible only once the rest of the paper has been digested. What is the 'appropriate calibration in ... time'? What is the 'single-sided' amplitude spectrum, and why is this feature significant? Would 'reveals' be better than 'retrieves'? What is the measurement rate (20kHz)? Why would the wavemeter be limited to a periodic signal?
p6 para 3 line 4	'wavelength meter reference which are depicted'

p6 para 3 line 5 'the readings agree well' – it's difficult for only one of them to agree
p6 para 3 line 5 omit 'signature'
p6 para 3 line 6 'retrieved to be 21 Hz', or perhaps 'determined to be 21 Hz'. To what precision?
p6 Fig 3 caption perhaps 'Wavelength measurement using a speckle wavemeter'
p7 para 1 line 7 10% of what? Modulation frequency? Wavelength??
p7 para 1 line 10 why is it important to note that a better wavemeter would have allowed more accurate absolute measurement?
p7 para 1 line 15 what is 'the resolution limit of the modulation signal'? Perhaps 'the limited resolution of the modulation signal source'?
p7 para 1 this paragraph is very long. Consider breaking at e.g. line 19 'A further experiment'
p8 Fig 4 caption 'b)' and 'c)' interchanged in final sentence
p8 para 2 line 1 'shows a wavelength'
p8 para 2 line 2 suggest 'with a 60 fm (30 MHz) wide'
p8 para 2 line 5 omit 'a'
p8 para 2 line 6 'perpetuation'?
p8 para 2 line 12 not really 'versatility'
p8 para 3 line 2 it would be helpful to distinguish between signal and spectral bandwidth
p9 para 2 line 2 is this an assertion, or has it been demonstrated? What is the role of the D-A converter in the wavemeter?
p9 para 2 line 7 omit 'window'
p10 para 2 line 1 what are the 'system sensitivities'? Is this intended to refer to limitations to the device performance?
p10 para 2 line 6 'capability by its sensitivity'
p10 para 2 line 10 is the laser linewidth really known to 1%?
p10 para 2 end the applications would seem to be much broader than the MOT laser stabilization described at length here
p10 para 3 line 2 'a standard method'
p11 para 1 line 2 does loaning equipment really get you co-authorship of a Nature paper nowadays?
references capitals in the article titles seem to appear as lower case
[17] journal title should be italic

Supplementary Methods

p2 para 1 line 3 wavelengths
p2 para 2 line 1 'wavelength variations of less than one nanometre'
p3 para 1 line 1 yield
p4 para 2 I am not convinced by the discussion of the turning point indicated by the arrow in Fig 2(a). Though no details are given of the 'training phase', I presume that the PCA process knows nothing of temporal variations but simply analyses the variations between individual speckle images. That there are turning points as functions of time is therefore irrelevant, and a valid interpretation of the curves of Fig. 2(a) and (b) would be to fold and map the 'speckle image' axis so as to represent the wavelength. There is no reason why PC1 should vary linearly with wavelength (consider a simple interference pattern: the fringes of a misaligned Michelson interferometer, similar to the ferris wheel example of [31]), and hence why its temporal dependence (or that of the higher components) should be harmonic. Indeed, in Fig 2(a), PC2-4 look distinctly anharmonic, as is apparent in their spectra of Fig 2(c). For small wavelength variations, however, PC1 is likely to be more linear and the higher components rather smaller, as is apparent in Figs 2(b), (d). The characteristic at the point indicated by the arrow is that it is a large wavelength excursion – that is, 'the wavelength variation and subsequent speckle variation', with respect to the mid-values, are ***greatest*** - not that it is a turning point as a function of time. It would be interesting to know whether the principal component compositions are the same in Figs 2(a) and (b) and, if so, the conditions under which they were deduced.
p5 para 1 line 10 lose
p5 para 2 line 3 'enhances ... decreased' tense?
p6 para 1 line 4 the calculation of speckle size, Nyquist frequency etc. is a useful one, and could be made more apparent in the text
p6 para 1 this paragraph is very long, and might be broken at, say, the end of line 13
p6 para 2 the English of this paragraph needs some work

p7 para 2 line 1 this first sentence is a fragment; perhaps 'While' should be omitted. The figure is Fig 2(b) of the main paper, not of the Methods.

p7 para 2 line 9 omit 'a'

p7 para 2 line 9 by 'for three different times', do the authors mean at three different times, or over three different periods? The caption to Fig 4 perhaps implies that 10 minutes was the duration, and that the measurements were at 0, 8 and 19 hours, whereas the text implies that the three values refer to the time at which the measurement was made.

p7 para 2 line 14 can the standard error deviation really be determined to three digits?

p7 para 2 the English of the second half of this paragraph needs some work

p7 para 3 line 1 'interaction'? 'propagation', perhaps

p7 para 3 line 7 'are'

p7 para 3 the English of this paragraph needs some work

p9 para 4 line 9 'resolution limit' usually refers to the smallest resolvable change; the text here refers instead to the largest change that can be measured linearly or unambiguously – ie the PCA measurement 'range'

[4], [16] journal titles should be italic

[25] year lacks brackets

Reviewer #2 (Remarks to the Author):

This paper is based on the well-established wavelength dependence of speckle in multiple scattering media, and it's a very clever and elegant application to laser stabilisation. I have no major comments and the following minor optional suggestions:

1. I think it would be good to reference the early work on the wavelength dependence of speckle, in particular George, N. & Jain, A. Appl. Phys. (1974) 4: 201.
2. In considering the stability of the system, might you also consider the angular dependence of the speckle?
3. Towards the end of the Supplementary methods, (p9) you claim that the "good agreement" ... actually the agreement is not great! means that the integrating there has "perfectly diffuse scatterers". I do not think that follows ... what one can say is that the resultant complex amplitude is a circular complex gaussian process for each polarisation component.

Response to Reviewer's comments:

We thank both referees for their positive response and insightful feedback on our work. We detail below our responses to their questions and suggestions.

Reviewer 1 comments and our reply.

Users will wish to know the wavelength range, resolution, sensitivity and update rate of the wavemeter, and its ability as a spectrometer to distinguish between simultaneous spectral components. The St Andrews group has demonstrated the resolution and update rate, and that the operating wavelength range is in principle as wide as that of the imaging array, sphere reflectivity, and fibre delivery. In its prototype implementation, however, only a narrow wavelength range has been covered continuously, and one wavelength at a time. That preliminary findings should be published before the full performance has been determined is correct and important; but it would be helpful if the paper were blunter in the abstract and introduction about what has and has not been demonstrated, and if perhaps it offered some speculation about these aspects in the later stages.

In order to summarise the current performance of the wavemeter, we have now included a specification table in Supplementary Note 1.

We have acknowledged that sub-fm resolution was only achieved in a narrow range near 780nm in the abstract,

"We demonstrate sub-femtometer resolution at 780nm and a calibrated measurement range from 488 to 1064nm"

and the penultimate paragraph in section 1

"... to yield wavelength resolution at the sub-femtometer level, which we demonstrate over a narrow wavelength range at 780nm."

The comments on the multiple spectral components being simultaneously measured have been reserved for the final sentence of Discussion.

"Finally, the extension of our approach to the simultaneous detection and measurement of multiple wavelengths [33] could also allow a single device to stabilize all of the various lasers (cooling, repumper, Raman and trapping) required in quantum technologies."

There were similarly a few places in which the language and terminology of the paper could have been clearer or more precise. Detection is not the same as measurement; nor is versatility the same as performance. Without further explanation, the meaning of bandwidth is unclear in a device for which both the measurement rate and optical spectral range are of interest. A wavelength below one nanometer is different from a wavelength variation of less than a nanometre. There are other places where one is left unsure whether a point has been demonstrated or just asserted; whether the transfer matrix method (TMM) and principal component analysis (PCA) were used separately or combined; and, if the latter, how this was done. A critical reading of the manuscript would identify a number of places at which the phrasing could be improved to render it more accurate and unambiguous. Perhaps because of the journal's unfortunate habit of separating methods from results and principles, there were also places where the logical order seemed a little muddled: I suggest that the authors restrict the methods section to unimportant technical details, and allow some of the important ones (such as of the integrating sphere, interference path lengths, imaging camera, and their relative geometry) into the main text.

Detection, versatility, bandwidth and wavelength vs wavelength variation have all been corrected as specified in the detailed comments below. The combined use of TMM and PCA has been addressed in point 31 below.

Extensive rephrasing and reordering has taken place throughout to improve the logical flow of the paper. In particular, we have now subdivided the Results section, and included the key information on the experiment and analysis.

Nonetheless, this is scientifically a compelling paper. The demonstration is utterly convincing, the characterization thorough (especially given that, when out-performing conventional instruments, it is not easy), and the illustrations clear and attractive. The 10 second duration of the temporal stability at the highest resolution (p10) might have been mentioned earlier – it is unsurprising in the apparent absence of thermal stabilization, and an aspect that is more interesting than limiting. I have a minor quibble, below, about the interpretation of Fig 2 of the Methods section, but the analysis is otherwise excellent.

The 10s duration of the lock is now included in the last paragraph of the introduction.

“We are able to stabilize an ECDL over 10s...”

The discussion of the turning point in the Methods figure has been removed from the manuscript, as detailed in point 64 below.

Aside from its technological and scientific implications, this work prompts one to wonder how speckle patterns evolve with wavelength, their topology (see e.g. Berry & Dennis, Proc R Soc Lond A 456, 2059 (2000)), whether their spatial frequency components alone are of interest, how the performance is enhanced by the two-dimensional measurement, how it will be affected when other spectral components are also present, and how the PCA capture range is determined by the apparatus parameters. While this is not the first paper to explore the phenomenon, the authors should not be shy to address, at least in passing, some of the more illuminating conceptual aspects and questions.*

This topic is well reviewed in the literature, and we include reference [1] as an example.

1. *title this could be snappier, and might for example mention the sub-fm performance*

We have modified the title from “Harnessing speckle patterns from an integrating sphere for a broadband, high resolution wavemeter and laser stabilization” to “Harnessing speckle from an integrating sphere for a sub-femtometer resolved broadband wavemeter and laser stabilization”.

2. *abstract, line 3 ‘original’ here is ambiguous; perhaps ‘novel’*

We deleted “original”.

“Our use of a fibre-coupled integrating sphere...”

3. *abstract, line 15 the laser is stabilized to the wavelength of the Rb line; there is no Rb in the control loop*

We agree, and removed “referenced to a Rubidium absorption line”

4. *p1 para 1 line 1 ‘in a disordered medium’, or ‘in disordered media’*

We changed “in a disordered media” to “in disordered media”.

5. *p2 para 1 line 1 ‘in such media’*

This sentence has been rephrased as “the speckle pattern that results from repeated scattering and interference is commonly deemed detrimental to an optical system.”

6. p2 para 1 line 3 can speckle patterns (as opposed to a scientific field or community) possess a paradigm? Indeed, is it so rare for a characteristic to be both a pro and a con?

This sentence has been combined with others and now reads “A coherent beam propagating in a disordered medium yields a unique speckle pattern that depends upon spatial and temporal characteristics of the incident light field” (p2 para 1 line 1)

7. p2 para 2 line 6 ‘a smaller footprint’; is ‘retain ... information’ really correct?

We removed “yet still retain an equivalent amount of information” (p2 para 1 line 5)

8. p2 para 3 line 4 perhaps replace ‘such as mode identification’ with e.g. ‘such as its transverse mode’

In refocusing the introduction, this sentence has been removed.

9. p2 para 4 line 2 ‘wavelength measurement’ or ‘wavelength determination’

This sentence was rewritten as “This acts like a highly sensitive and complex interferometer due to internal path-length differences on the meter scale, delivering a different speckle pattern for each wavelength...” (p2 para 2 line 2)

10. p2 para 4 line 5 it would be interesting to learn how ‘exceptionally long’ is the interfering path difference

The mean path is $Z_0 = 4R/3(1-r)$ where R is the sphere inner radius and r the reflectivity of the surface (99% for spectralon) which gives a path length of 3.34m for $R=25\text{mm}$. The Probability Density Function for the trajectories from an integrating sphere can be approximated by $\text{PDF} = C \cdot r^{z/z_0}$ [with C a normalisation constant such that the integral from 0 to infinity =1] meaning that the $1/e^2$ width of the distribution is $\sim 2Z_0$ for our parameters, i.e. 6.64m. For a sphere with exit port radius = $0.1R$, and all light exiting the port captured by the CCD, the minimum path is $2 \cdot r + 4/3 \cdot r + 3/4 \cdot L$, which is $\sim 20\text{cm}$. Therefore we have included “meter scale” in the sentence highlighted in point 9 above.

11. p2 para 4 line 8 ‘two orders of magnitude better’

We replace “two orders of magnitude greater” by “two orders of magnitude better”.

12. p2 para 4 line 10 it would be helpful to distinguish throughout between signal bandwidth, (optical) spectral bandwidth and spectral operating range

We have used “range” or “spectral operating range” instead of bandwidth throughout the manuscript

13. p2 para 4 line 11 ‘visible and near-infrared spectrum’

We change from “visible” to “visible and near-infrared” (p2 para 2 line 8)

14. p2 para 5 line 2 ‘laser frequency fluctuations’

This sentence has been deleted in the rewrite.

15. *p2 para 5 line 3 'use of appropriate electronic feedback'*

We modify from "appropriate feedback" to "appropriate electronic feedback". (p2 para 3 line 1)

16. *p2 para 5 line 6 'absorption, as are Fabry-Perot based'*

We change from "similar to fabry-perot..." to "as with Fabry-Perot..." (p2 para 3 line 7)

17. *p2 para 5 line 7 'the capture range...' sentence is partly repeated; omit first version*

In the reordering, this information has been removed from the introduction and can now be found in Results – Laser Stabilization – p7 para 3 line 8 – and Discussion – p9 para3 line 7.

18. *p3 para 1 line 2 omit comma after 'sphere'*

This sentence was changed substantially in the new subsection "Wavemeter Setup" which combined parts of the old Supplementary Methods and Results sections.

19. *p3 para 1 line 4 were the lasers really evaluated, or simply used?*

We modify "evaluated" to "employed". This sentence was moved to Supplementary Information page 2, para 1, line 2.

20. *p3 para 1 this paragraph is long and a little jumbled; it contains many technical details (e.g.ECDL, F-P interferometer, frame grabber, computer processor) that are irrelevant to the thrust of the paper and would sit better in the 'methods' section, but omits the characteristics of the integrating sphere.*

This paragraph has been rewritten, with some information moving to Supplementary Note S2, and the rest forming the aforementioned subsection "Wavemeter Setup"

21. *p3 para 1 line 31 given that the speckle pattern results from the interference between different 'trajectories', perhaps some hint of this process could be given here, and the 'trajectory' description emphasized less?*

The trajectory description has been removed.

22. *p3 para 1 line 37 'one would require a 300 km long fibre' (note that the spelling of 'fibre'/'fiber' is inconsistent in this paper)*

We change all "fiber" to "fibre" to be consistent on spelling

23. *p4 para 1 line 6 what is meant by 'a high degree of disorder'? Has this been quantified/measured?*

By 'a high degree of disorder' we mean that the speckle pattern is extremely sensitive to any change of the input parameters or the geometry of the measurement device itself. This makes it impossible to predict speckle change with respect to the change of any input parameters but the direct calibration has to be performed. In the process of reordering these paragraphs, this sentence was

deemed unnecessary and removed.

24. *p4 para 1 line 8 one could begin a new paragraph at 'The core of...'*

We have started a new subsection at "The core of..." (p3 para 3 line 1)

25. *p4 para 1 line 10 omit 'out'*

This sentence was rewritten to avoid the ambiguity discussed in point 31.

26. *p4 para 1 line 14 what is meant by 'wavelength determination above one nanometer'? Perhaps 'wavelength determination for variations above one nanometer'?*

We modify this paragraph by deleting "to determine the laser wavelength with a nanometer resolution." and change "for wavelength determination above one nanometer" to "for wavelength determination with a nanometer resolution". (p3 para 3 line 4)

27. *p4 para 1 line 17 the calibration process would benefit from further explanation. The laser sources are perhaps irrelevant: several common laser lines were used.*

This has been changed to "The intensity distribution recorded by the camera is $I = T S$, where S is the input spectrum and the transmission matrix T . This is constructed in a calibration stage, from speckle patterns recorded using established and well-known laser sources across the required wavelength range with a step of 1 nm ." (p3 para 3 line 7)

28. *p4 para 1 line 21 the 'truncated singular value decomposition' might deserve a reference for the uninitiated*

We add the reference to Redding *et al.* Opt. Express 21, 6584-6600 (2013). (p5 para 1 line 2)

29. *p4 para 1 line 29 is the operational range limited to 488-1064 nm, or is this just the range over which it has been demonstrated?*

We modify the sentence to "The demonstrated operational range of ..." (p5 para 1 line 8)

30. *p5 Fig 2 caption it would be helpful to explain the abbreviations TMM, PCA and ECDL within the caption. The resolution demonstrated in (b) appears to be rather better than the 1 nm stated.*

We add description of abbreviations in the figure caption: "transmission matrix method (TMM)", "principle component analysis (PCA)" and "extended cavity diode laser (ECDL)"

31. *p5 para 1 line 8 are the PCA and TM methods used together?*

In the present study TM and PCA methods are used independently as the PCA calibration was usually used over a single spectral range. However, to measure a completely unknown wavelength at sub-nanometer precision, one would use these methods together in a sequence: TM first for the determination of the spectral position within 1 nm precision and then PCA for precision measurements better than 1 nm. To make this clearer, we have rephrased the opening to each paragraph in the subsection Detection Algorithm.

"... To achieve a broad operating range, a coarse analysis utilizing the transmission matrix method..."

“... To achieve a high resolution we utilize a different classification algorithm capable of accurately distinguishing small speckle changes cause by wavelength changes below 1nm. This algorithm is based on principal component analysis...”

and in the penultimate sentence of the subsection

“We note that, in order to measure an unknown wavelength to high precision, these two algorithms can be used sequentially.”

32. p5 para 1 line 9 perhaps ‘Principal component analysis (or singular value decomposition (SVD)) is an established tool’, or ‘Principal component analysis and singular value decomposition (SVD) are established tools’

We modify our sentence: from “Principal component analysis or singular value decomposition (SVD) are established tools” to “Principal component analysis (or singular value decomposition (SVD)) is an established tool...”. (page 5 para 2 line 5)

33. p6 para 1 it would be interesting to know how the wavelength accuracy varies with its proximity to the calibration wavelengths

For the PCA measurement the measured wavelength needs to be within the calibration range and the accuracy is almost constant across the whole range. If the wavelength goes outside the calibration range or the system capture range is insufficient then PCA cannot retrieve the wavelength correctly. The dependence of PCA accuracy on the proximity to calibration wavelength was previously investigated in [16], and will be investigated for our fm-resolved wavemeter in future work.

34. p6 para 2 this paragraph is unclear, and becomes partly intelligible only once the rest of the paper has been digested. What is the ‘appropriate calibration in ... time’? What is the ‘single-sided’ amplitude spectrum, and why is this feature significant? Would ‘reveals’ be better than ‘retrieves’? What is the measurement rate (20kHz)? Why would the wavemeter be limited to a periodic signal?

“The acquisition rate of the wavemeter depends on the camera speed and can be above 200 kHz for a small detector area.” has been added to the Results (p3 para 2 line 7).

Due to a previous referee’s comment that our device can only measure periodic signals we had included this clarification in the main text, but this has now been removed.

We agree that the inclusion of the single-sided amplitude spectrum for these measurements is a little superfluous, and these have been removed from Figure 3.

Otherwise, the paragraph has been rewritten for clarity (p5 para 3 – p7 para 2)

35. p6 para 3 line 4 ‘wavelength meter reference which are depicted’

This has been rewritten as “Fig 3a) shows the good agreement between...” (p5 para 4 line 1)

36. p6 para 3 line 5 ‘the readings agree well’ – it’s difficult for only one of them to agree

As in point 35

37. p6 para 3 line 5 omit ‘signature’

The single-sided amplitude spectra have been removed from this section (see point 34)

38. p6 para 3 line 6 'retrieved to be 21 Hz', or perhaps 'determined to be 21 Hz'. To what precision?

The single-sided amplitude spectra have been removed from this section (see point 34)

39. p6 Fig 3 caption perhaps 'Wavelength measurement using a speckle wavemeter'

We modify from "Wavelength measurement of a speckle wavemeter" to "Wavelength measurement using a speckle wavemeter"

40. p7 para 1 line 7 10% of what? Modulation frequency? Wavelength??

We clarify this by changing "... each other." to "within 10% of interpolated Fabry-Perot interferometer measurements of the wavelength." (page 5 para 4 line 7)

41. p7 para 1 line 10 why is it important to note that a better wavemeter would have allowed more accurate absolute measurement?

We would like to emphasize that due to the accuracy limitation of the commercial wavemeter we cannot provide comparison of the absolute measurements between the two. We have rephrased this as "Clearly, with a ..." (p6 para 1 line 4)

42. p7 para 1 line 15 what is 'the resolution limit of the modulation signal'? Perhaps 'the limited resolution of the modulation signal source'?

We modify our sentence from "the resolution limit of the modulation signal" to "the limited resolution of the modulation signal source". (p6 para 1 line 7)

43. p7 para 1 this paragraph is very long. Consider breaking at e.g. line 19 'A further experiment'

We insert a paragraph break(p6 para 1 – 2), and have rewritten the start of the new paragraph.

44. p8 Fig 4 caption 'b)' and 'c)' interchanged in final sentence

We correct caption labelling.

45. p8 para 2 line 1 'shows a wavelength'

We insert article "a". (p7 para 4 line 1)

46. p8 para 2 line 2 suggest 'with a 60 fm (30 MHz) wide'

As the relationship between fm and MHz is given in p8 para 1 line 2, we omit the frequency in p7 para 4 line 2-3.

47. p8 para 2 line 5 omit 'a'

We omit "a". (p7 para 4 line 5)

48. p8 para 2 line 6 'perpetuation'?

This paragraph (p7 para 5) has been rewritten for clarity.

49. *p8 para 2 line 12 not really 'versatility'*

This sentence was removed in the rewrite (see point 48)

50. *p8 para 3 line 2 it would be helpful to distinguish between signal and spectral bandwidth*

We have changed from "broad operating bandwidth" to "broad operating range". (p8 para 1 line 2)

51. *p9 para 2 line 2 is this an assertion, or has it been demonstrated? What is the role of the D-A converter in the wavemeter?*

We remove the word "solely" (p9 para 1 line 3) to affirm that, at present, the limited resolution of our amplitude modulation source is the first obstacle to be overcome.

The D-A converter limits the resolution of the amplitude modulation signal output from our National Instruments card to the current control of the ECDL.

"Progress towards attometer-level resolution using this approach is currently limited by our 16-bit amplitude modulation source and the availability of an accurate wavelength reference to calibrate the variability of the principal components." (p9 para 1 line 3)

52. *p9 para 2 line 7 omit 'window'*

We omit "window". (p9 para 1 line 9)

53. *p10 para 2 line 1 what are the 'system sensitivities'? Is this intended to refer to limitations to the device performance?*

In order to clarify this point we modify this paragraph to "Our present system is limited in its long-term stability (see Supplementary Note S6), but mechanical and thermal stabilization engineering could form part of a future study. Alternatively, a regular recalibration step, as can be found in many commercial wavemeters, can be used to maintain accuracy over longer periods of time." (p9 para 3)

54. *p10 para 2 line 6 'capability by its sensitivity'*

as in point 53

55. *p10 para 2 line 10 is the laser linewidth really known to 1%?*

This is now quoted to 2 significant figures (p7 para 4 line 6)

56. *p10 para 2 end the applications would seem to be much broader than the MOT laser stabilization described at length here*

We have added a longer discussion on the applicability of the method to quantum technologies based on cold atoms (p9 para 4)

57. *p10 para 3 line 2 'a standard method'*

This sentence is modified to "Furthermore, compared to dither locking of the laser to an atomic saturated absorption feature, speckle-based stabilization has the advantage of eliminating any frequency modulation of the laser" (p9 para 4 line 5)

58. *p11 para 1 line 2 does loaning equipment really get you co-authorship of a Nature paper nowadays?*

59. *references capitals in the article titles seem to appear as lower case*

We have corrected this

60. *[17] journal title should be italic*

We correct journal title to italic.

Supplementary Methods

61. *p2 para 1 line 3 wavelengths*

We modify “wavelength” to “wavelengths” (p2 para 4 line 4)

62. *p2 para 2 line 1 ‘wavelength variations of less than one nanometre’*

We modify “wavelength below one nanometer” to “wavelength variations of less than one nanometer”. (p3 para 2 line 1)

63. *p3 para 1 line 1 yield*

This sentence has been removed in the rewrite.

64. *p4 para 2 I am not convinced by the discussion of the turning point indicated by the arrow in Fig 2(a). Though no details are given of the ‘training phase’, I presume that the PCA process knows nothing of temporal variations but simply analyses the variations between individual speckle images. That there are turning points as functions of time is therefore irrelevant, and a valid interpretation of the curves of Fig. 2(a) and (b) would be to fold and map the ‘speckle image’ axis so as to represent the wavelength. There is no reason why PC1 should vary linearly with wavelength (consider a simple interference pattern: the fringes of a misaligned Michelson interferometer, similar to the ferris wheel example of [31]), and hence why its temporal dependence (or that of the higher components) should be harmonic. Indeed, in Fig 2(a), PC2-4 look distinctly anharmonic, as is apparent in their spectra of Fig 2(c). For small wavelength variations, however, PC1 is likely to be more linear and the higher components rather smaller, as is apparent in Figs 2(b), (d). The characteristic at the point indicated by the arrow is that it is a large wavelength excursion – that is, ‘the wavelength variation and subsequent speckle variation’, with respect to the mid-values, are **greatest** - not that it is a turning point as a function of time. It would be interesting to know whether the principal component compositions are the same in Figs 2(a) and (b) and, if so, the conditions under which they were deduced.*

We agree that PCA knows nothing about time variation. This discussion has been removed from the manuscript.

65. *p5 para 1 line 10 lose.*

This sentence removed in the rewrite

66. *p5 para 2 line 3 ‘enhances ... decreased’ tense?*

We change to present from “decreased” to “decreases” (p6 para 2 line 6)

67. *p6 para 1 line 4 the calculation of speckle size, Nyquist frequency etc. is a useful one, and could be made more apparent in the text*

We have added a reference to this discussion in the main text (p3 para 2 line 5)

68. *p6 para 1 this paragraph is very long, and might be broken at, say, the end of line 13*

We introduce a paragraph at starting from “Additionally to enhance the spatial...” (p7 para 1)

69. *p6 para 2 the English of this paragraph needs some work*

We have substantially modified this paragraph. (p7 para 2)

70. *p7 para 2 line 1 this first sentence is a fragment; perhaps ‘While’ should be omitted. The figure is*

We modify “While Fig. 2 b) inset” to “The inset of Fig. 2 b) of the main paper” (p7 para 3 line 1)

71. *Fig 2(b) of the main paper, not of the Methods.*

as in point 70

72. *p7 para 2 line 9 omit ‘a’*

This section has been rewritten, superseding the change

73. *p7 para 2 line 9 by ‘for three different times’, do the authors mean at three different times, or over three different periods? The caption to Fig 4 perhaps implies that 10 minutes was the duration, and that the measurements were at 0, 8 and 19 hours, whereas the text implies that the three values refer to the time at which the measurement was made.*

The measurement was performed at three different times. We correct the text from “for three different times” to “**at three different times**” (p7 para 2 line 9) and Fig. 4 caption: from “over three different times” to “**at three different times**”.

74. *p7 para 2 line 14 can the standard error deviation really be determined to three digits?*

Since the large part of this error is due to the systematic shift, we keep the same lowest significant digit in all the measurements. After the correction of the systematic shift we have only two significant digits left. We also correct the standard error after the correction factor was applied in the main text from “60 fm” to “82 fm”, the same number as described in the Figure 9 caption.

75. *p7 para 2 the English of the second half of this paragraph needs some work*

We have modified this part of the paragraph to improve the readability. (p7 para 3)

76. *p7 para 3 line 1 ‘interaction’? ‘propagation’, perhaps*

We correct “random interaction” to “**propagation**” (p8 para 3 line 1)

77. p7 para 3 line 7 'are'

We correct "is" to "are" (p9 para 1 line 2)

78. p7 para 3 *the English of this paragraph needs some work*

We re-wrote this paragraph. (p8 para 3)

79. p9 para 4 line 9 'resolution limit' usually refers to the smallest resolvable change; the text here refers instead to the largest change that can be measured linearly or unambiguously – ie the PCA measurement 'range'

We change to "operating range" (p10 para 1 line 6)

80. [4], [16] journal titles should be italic

We correct journal titles to italic

81. [25] year lacks brackets

We add brackets to the publishing year.

Reviewer 2 comments and our reply.

1. *I think it would be good to reference the early work on the wavelength dependence of speckle, in particular George, N. & Jain, A. Appl. Phys. (1974) 4: 201.*

We add the citation to this article in the introduction. (p2 para 1 line 3)

2. *In considering the stability of the system, might you also consider the angular dependence of the speckle?*

Increasing the lock stability beyond 10 seconds will be the basis of future study, for which we will consider both temperature stabilization and using the angular dependence of the speckle to give us more information.

3. *Towards the end of the Supplementary methods, (p9) you claim that the "good agreement" ... actually the agreement is not great! means that the integrating there has "perfectly diffuse scatterers". I do not think that follows ... what one can say is that the resultant complex amplitude is a circular complex gaussian process for each polarisation component.*

We agree with the referee's comment, and have therefore removed reference to "perfectly diffuse scatterers" and replaced with

"The experimental and simulated speckle pattern agree qualitatively, and indicate that the resultant complex amplitude is a circular complex Gaussian process [31,32]" (p9 para 3)

REVIEWERS' COMMENTS:

Reviewer #1 (Remarks to the Author):

I'm grateful to the authors for dealing so attentively with the reviewers' concerns. The detailed and careful rebuttal makes the revised document very straightforward to consider.

In addition to satisfactorily addressing the points made by the article's reviewers, the authors have significantly improved the clarity and structure of the article to produce an excellent paper that is far clearer and easier to read. I remain firmly of the view that this is exciting, significant and beautiful work that in all probability will result in a change of paradigm for precision spectroscopy, both in the lab and in technological applications. The manuscript now does the work full justice!

I am pleased to recommend it for publication in Nature Communications.

Reviewer #2 (Remarks to the Author):

This revised manuscript addresses the very minor concerns I had.